# Burden of acute lymphoblastic leukemia in children and adolescents in low- and middle-income countries from 1990 to 2023 and projections to 2050: A systematic analysis from the global burden of disease study 2023

Peng Liu[1,2], ZiXin Xu[1,2], Wenfu Song[1], Jianxiong Yang[1], Jianbao Li[1,2,3]*

**1** Hospital of Chengdu University of Traditional Chinese Medicine, Chengdu, China, **2** Chengdu University of Traditional Chinese Medicine, Chengdu, China, **3** Departmalest of Respiratory Medicine, Hospital of Chengdu University of Traditional Chinese Medicine, Chengdu, China

* Lijianbao@cdutcm.edu.cn

## Abstract

### Background

Acute Lymphoblastic Leukemia (ALL) is the most common and curable malignancy in children and adolescents, yet it remains a significant health threat. Despite global survival improvements driven by new drugs and treatment protocols, comprehensive data on disease burdens and attributable risk factors across diverse socio-economic contexts remain limited. This study evaluated the epidemiological characteristics, temporal trends, and modeled attributable risk factors of ALL in LMICs from 1990 to 2023.

### Methods

Using GBD 2023 data, we analyzed ALL incidence, mortality, and DALYs in children and adolescents (0–19 years), stratified by World Bank income groups. We assessed temporal trends via Joinpoint regression, projected burdens to 2050 using the Bayesian Age-Period-Cohort (BAPC) model, and evaluated risk attribution for occupational benzene and formaldehyde exposure.

### Results

In 2023, LMICs reported 64,477 new ALL cases and 30,909 deaths, with a higher burden in males. While the age-standardized incidence rate (ASIR) declined overall, absolute cases rose due to population growth. Although occupational benzene and formaldehyde exposure were the only modeled risk factors, their Population Attributable Fractions (PAFs) were negligible (<1%), indicating minimal contribution to the total burden. Projections suggest the age-standardized ALL burden will

**Data availability statement:** All data used in this study are publicly available from the Global Burden of Disease (GBD) 2023 study. The data on incidence, mortality, and DALYs for ALL, along with their 95% uncertainty intervals, were obtained from the GBD Results Tool with a registered account (https://vizhub.healthdata.org/gbd-results/). The underlying sources of data are accessible via the GBD 2023 Sources Tool (https://ghdx.healthdata.org/gbd-2023/sources). A detailed description of data sources and extraction methods is provided in the Methods section of this manuscript. All data generated or analyzed during this study are included in this published article and its supplementary information files.

**Funding:** The author(s) received no specific funding for this work.

**Competing interests:** The authors have no conflicts of interest to disclose.

continue declining through 2050, with the sharpest decrease expected in low-income countries.

## Conclusion

Globally, childhood and adolescent ALL mortality and DALYs are declining, yet the burden remains substantial in many LMICs, with the smallest improvements observed in low-income countries from 1990 to 2023. Occupational benzene and formaldehyde exposure contributed minimally to the total burden (PAFs < 1%), indicating that direct occupational exposure is rare in children. These findings suggest that the persistent burden in LMICs is primarily driven by healthcare system factors rather than occupational environmental exposures. Therefore, efforts to further reduce ALL mortality and DALYs in high-burden regions should address gaps in healthcare access and treatment delivery.

## Introduction

Acute lymphoblastic leukemia (ALL) is the most common and highly treatable cancer in children and adolescents. In recent years, the overall incidence of ALL has decreased, with about 60% of cases diagnosed before the age of 20. As the most common childhood cancer worldwide, ALL accounts for about 25% of all cancer diagnoses in children under 15. At diagnosis, lymphoblasts in various stages of development proliferate abnormally, infiltrate the bone marrow, suppress normal blood cell production, and spread to multiple organs [1]. While highly treatable, outcomes vary significantly: 5-year survival rates exceed 90% in high-income countries but remain substantially lower in low- and middle-income countries (LMICs)—a disparity driven by limited healthcare resources, delayed diagnosis, and poor access to consistent, high-quality treatment [2].

Thanks to advances in drug development and improvements in treatment protocols, the treatment outcomes for pediatric and adolescent ALL have improved significantly through large clinical trials in recent decades [3]. However, significant disparities remain: while the 5-year survival rate exceeds 90% in high-income countries, it remains much lower in low- and middle-income countries (LMICs). This is mainly due to limited healthcare resources, delayed diagnosis, poor access to treatment, and heavy financial burdens. Additionally, despite improvements in survival rates, about 25% of patients relapse, and around 10% fail to achieve a complete cure, making acute lymphoblastic leukemia (ALL) one of the leading causes of cancer-related deaths in children. The disease and its intensive treatments can also cause long-term side effects in survivors, such as neurocognitive impairments, endocrine dysfunction, cardiotoxicity, and secondary cancers [4].

Although incidence and survival differ across regions and over time, few studies have used comparable population-based data to quantify how changes in modifiable risk factors contribute to these trends [5]. This study aims to assess the global burden of pediatric and adolescent ALL from 1990 to 2023, with a focus on LMICs, and to evaluate

the contribution of key risk factors—including occupational benzene and formaldehyde exposure [6]. By linking disease burden to specific, actionable exposures, our findings can help prioritize public health strategies where they are most needed [7].

## Methods

### Study design and data sources

This study utilized data from the Global Burden of Disease Study 2023 (GBD 2023) to analyze the burden of Acute Lymphoblastic Leukemia (ALL) in children and adolescents aged 0–19 years [8]. We focused specifically on Low- and Middle-Income Countries (LMICs), categorized according to the World Bank classification into low-income (LICs), lower-middle-income, and upper-middle-income (UMICs) countries.

Data on incidence, mortality, and Disability-Adjusted Life Years (DALYs) for ALL, along with their 95% Uncertainty Intervals (UIs), were obtained from the GBD Results Tool (https://vizhub.healthdata.org/gbd-results/). The GBD 2023 methodology for estimating cancer burden involves a systematic analysis of vital registration, censuses, surveys, and cancer registry data, which has been described in detail elsewhere. All data used in this study are publicly available through the GBD 2023 Sources Tool (https://ghdx.healthdata.org/gbd-2023/sources).

### Outcome measures and standardization

The primary outcomes included incidence, mortality, and DALYs. DALYs are calculated as the sum of Years of Life Lost (YLLs) due to premature mortality and Years Lived with Disability (YLDs). YLLs are calculated by multiplying the number of deaths at each age by the standard life expectancy at that age. YLDs are calculated by multiplying the prevalence of the sequelae (health consequence) by the disability weight for that sequela.

To facilitate comparison across populations with different age structures, we used Age-Standardized Rates (ASRs). These rates were calculated by applying the age-specific rates of each location to a standard GBD global reference population. This allows us to report the rates per 100,000 population while removing the confounding effect of demographic changes over time.

### Attributable burden analysis

The study of attributable burden follows the comparative risk assessment framework used in the GBD [8]. Within this framework, only specific risk factors with sufficient epidemiological evidence are modeled as causes of disease burden. In this study, we included two risk factors that are modeled as being attributable to ALL DALYs in the GBD framework: occupational exposure to formaldehyde and occupational exposure to benzene. These two risk factors were selected because they are the only occupational chemical exposures with sufficient epidemiological evidence to be included in the GBD comparative risk assessment framework for ALL, and they represent modifiable exposures of potential public health relevance in LMICs settings.

For each risk-outcome pair, the Population Attributable Fraction (PAF) was calculated to estimate the proportional reduction in disease burden that would occur if exposure to a risk factor were reduced to an ideal theoretical minimum exposure level. The general formula for PAF is as follows:

$$PAF = \frac{\sum_{i=1}^{k} P_i (RR_i - 1)}{\sum_{i=1}^{k} (P_i \times RR_i)}$$

### Statistical analysis

Temporal trends in age-standardized incidence, mortality, and DALY rates from 1990 to 2023 were assessed using Joinpoint regression analysis (version 5.4.0; National Cancer Institute). ln(rate)=β×year+ϵ was fitted to the data to estimate the

Annual Percent Change (APC) and corresponding 95% Confidence Intervals (CIs) for each identified trend segment. The Average Annual Percent Change (AAPC) was calculated to summarize overall trends across the entire study period.

To project future trends in the burden of ALL through 2050, we employed a Bayesian Age-Period-Cohort (BAPC) model. It is important to note that BAPC models assume the continuity of historical trends and may not fully capture sudden structural changes in healthcare surveillance or environmental regulations, particularly in dynamic LMICs settings. Therefore, all projections are presented with 95% UIs to reflect these uncertainties. Data visualization and figure generation were performed using R software (version 4.5.1). The statistical code used for the GBD 2023 analysis is publicly available online.

### Ethics statement

This study involves a secondary analysis of publicly available, anonymized data from the Global Burden of Disease Study 2023 (GBD 2023). The dataset contains no personally identifiable information, with all estimates aggregated at the national or subnational level. Given that the research involved no direct interaction with human participants and utilized only de-identified data, informed consent was not required. Furthermore, ethical approval was not necessary in accordance with international guidelines governing the use of publicly released, anonymized epidemiological data.

## Results

### Burden of ALL in 2023

In 2023, low- and middle-income countries (LMICs) recorded 64,477 new ALL cases (95% UI: 48,971–79,983) among children and adolescents aged 0–19 years. Males accounted for 37,153 cases, while females accounted for 27,324 cases. Total deaths numbered 30,909 (95% UI: 26,075–35,742), with a higher burden in males (18,383) than in females (12,526). Total DALYs reached 2,532,465 (95% UI: 2,134,526–2,930,403), similarly dominated by males (1,506,053 vs. 1,026,411 in females). At the national level, Mozambique had the highest age-standardized incidence rate (ASIR: 7.7 per 100,000), followed by Micronesia and Liberia. Ethiopia had the highest age-standardized mortality rate (ASMR: 4.5 per 100,000), while Cuba had the highest age-standardized DALY rate (ASDR: 99.3 per 100,000) (S1 Table).

### Burden by income group and age in 2023

Low-income countries (LICs):For both sexes combined, ASIR was 3.02 per 100,000 (male: 3.53; female: 2.49). ASMR was 2.01 (male: 2.36; female: 1.65), and ASDR was 164.65 (male: 193.96; female: 134.48). The highest ASIR, ASMR, and ASDR were observed in males aged 2–4 years and females aged <1 year, indicating distinct age and sex patterns in the most resource-limited settings. Lower-middle-income countries (LMCs):ASIR was 1.63 per 100,000 (male: 1.87; female: 1.37). ASMR was 0.96 (male: 1.12; female: 0.80), and ASDR was 79.24 (male: 92.20; female: 65.57). The highest ASIR in males occurred at age 2–4 years; females showed similar rates in the < 1 year and 2–4 years groups. The highest ASMR and ASDR for both sexes were observed in the 2–4 years group, highlighting this age as the most vulnerable in lower-middle-income settings. Upper-middle-income countries (UMICs):ASIR was 4.37 per 100,000 (male: 4.72; female: 3.99). ASMR was 1.53 (male: 1.75; female: 1.30), and ASDR was 125.11 (male: 141.88; female: 106.91). Age-specific patterns differed notably from other income groups: ASIR peaked in infancy (<1 year) for both sexes. ASMR and ASDR peaked in males aged 2–4 years, whereas in females, they peaked in the < 1 year group, suggesting sex-specific vulnerabilities in early life (Fig 1; S2 Table).

### Temporal trends from 1990 to 2023

From 1990 to 2023, the absolute number of new ALL cases in LMICs declined by 17.6% (from 78,197–64,477), deaths by 40.2% (from 51,703–30,909), and DALYs by 40.6% (from 4,266,646–2,532,465). Age-standardized rates also decreased

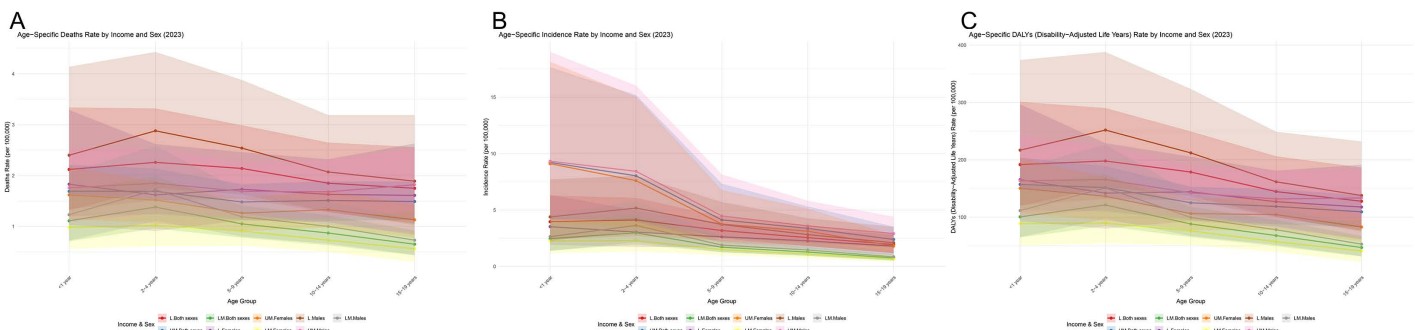

**Fig 1. Age-specific rates of acute lymphoblastic leukemia in low- and middle-income countries in 2023.** (A) incidence rate, (B) mortality rate, and (C) DALY rate.

across all income groups, with the magnitude of decline increasing by national income level (S3 Table; Table 1). LICs: ASIR (AAPC –1.12), ASMR (AAPC –1.34), and ASDR (AAPC –1.37) all declined significantly, with steeper reductions in females than males. The largest improvements were observed in the 5–9 years age group. LMCs: Similar declines were observed for ASIR (AAPC –1.26), ASMR (AAPC –1.73), and ASDR (AAPC –1.75), again most pronounced in children aged 5–9 years. Females experienced larger reductions than males across all three metrics. UMICs: ASIR showed a modest decline (AAPC –0.79), while ASMR (AAPC –2.58) and ASDR (AAPC –2.60) fell sharply, especially in infants (<1 year), reflecting substantial improvements in treatment and care in higher-resource settings within LMCs.

## Attributable DALYs for ALL in 2023

In 2023, risk factor exposure was estimated to cause 2,251 (95% UI: 1,190–3,311) DALYs attributable to ALL in LMICs. Table 2 presents the contributions of two specific risk factors across both sexes and the three World Bank income groups. Occupational exposure to benzene was the leading among the assessed risk factors for ALL DALYs in both sexes across all income levels, particularly among females in low-income countries (LICs) and upper-middle-income countries (UMICs). Occupational exposure to formaldehyde ranked second, with nearly equal contributions observed in males and females. Notably, UMICs bore the highest burden from both risk factors compared with the other two income groups (Table 2; S4 Table).

## The trend of risk factors of ALL burden from 1990 to 2023

The number of DALYs attributable to risk factors increased from 407 (95% UI: 143–703) in 1990 to 898 (95% UI: 315–1,585) in 2023. Only occupational benzene and formaldehyde exposure were modeled as attributable risk factors for ALL within the GBD framework. In 2023, they accounted for only 2,251 DALYs (95% UI: 1,190–3,311) in LMICs, representing a negligible fraction (PAF <1%) of the total ALL burden. UMICs bore the highest absolute burden from both risk factors, reflecting larger industrial workforces, while LICs had the lowest attributable burden (Table 2; S4 Table). From 1990 to 2023, age-standardized DALY rates attributable to these exposures remained statistically stable across all income groups, with no significant annual percent change (all 95% UIs included zero) S5 Table.

## Projected ALL burden to 2050

We projected the age-standardized incidence rate (ASIR) and age-standardized mortality rate (ASMR) for acute lymphoblastic leukemia (ALL) among children and adolescents in low- and middle-income countries (LMICs) from 2023 to 2050 using Bayesian age-period-cohort (BAPC) models. From 2023 to 2050, the ASIR and ASMR of ALL in this population are

**Table 1. Average annual percent change in incidence, mortality, and DALYs rates for different age groups and World Bank Income.**

| Age group | World Bank Low Income | | | World Bank Lower Middle Income | | | World Bank Upper Middle Income | | |
|---|---|---|---|---|---|---|---|---|---|
| | AAPC of incidence rate, % (95% CI) | AAPC of mortality rate, % (95% CI) | AAPC DALYs rate, % (95% CI) | AAPC of incidence rate, % (95% CI) | AAPC of mortality rate, % (95% CI) | AAPC DALYs rate, % (95% CI) | AAPC of incidence rate, % (95% CI) | AAPC of mortality rate, % (95% CI) | AAPC DALYs rate, % (95% CI) |
| <1 year | −0.82 (−1.06 to −0.57) | −1.2 (−1.46 to −0.94) | −1.2 (−1.46to −0.95) | −0.38 (−0.60 to −0.16) | −0.65 (−0.84 to −0.46) | −1.22 (−1.51 to −0.93) | −2.45 (−2.58 to −2.32) | −5.86 (−6.07 to −5.64) | −5.77 (−5.98 to −5.56) |
| 2-4 years | −0.9 (−1.21 to −.6) | −1.28 (−1.59 to −0.97) | −1.28 (−1.59 to −0.97) | −0.95 (−1.28 to −0.61) | −0.77 (−0.97 to −0.56) | −1.7 (−1.97 to −1.43) | −0.01 (−0.08 to −0.07) | −2.95 (−3.08 to −2.81) | −2.88 (−3.01to −2.75) |
| 5-9 years | −1.59 (−1.74 to −1.43) | −1.81 (−1.96 to −1.65) | −1.81 (−1.97 to −1.66) | −1.63 (−1.76 to −1.49) | −0.79 (−0.96 to −0.63) | −2.03 (−2.13 to −1.93) | −1.63 (−1.76 to −1.49) | −2.93 (−3.02 to −2.85) | −2.9 (−2.98 to −2.81) |
| 10-14 years | −1.16 (−1.3 to −1.02) | −1.35 (−1.49 to −1.21) | −1.35 (−1.49 to −1.21) | −1.05 (−1.19 to −0.92) | −0.76 (−0.88 to −0.64) | −1.37 (−1.48 to −1.25) | −0.81 (−0.99 to −0.62) | −2.26 (−2.41 to −2.12) | −2.25 (−2.39 to −2.1) |
| 15-19 years | −1.25 (−1.38 to −1.13) | −1.4 (−1.52 to −1.27) | −1.4 (−1.52 to −1.27) | −1.07 (−1.16 to −0.97) | −0.71 (−0.80 to −0.63) | −1.3 (−1.4 to −1.19) | −1.27 (−1.52 to −1.03) | −2.4 (−2.61 to −2.19) | −2.38 (−2.6 to −2.17) |

AAPC, average annual percent change; CI, confidence interval; DALYs, disability-adjusted life-years.

**Table 2. Percentage contribution of risk factors to <20 years DALYs of ALL in 2023, for sexes and World Bank Income.**

| Location | Sex | Risk factors | |
|---|---|---|---|
| | | Occupational exposure to benzene, % (95%UI) | Occupational exposure to formaldehyde, % (95%UI) |
| World Bank Low Income | Both sexes | 0.08 (0.13-0.02) | 0.02 (0.04-0.01) |
| World Bank Low Income | Male | 0.06 (0.11-0.02) | 0.02 (0.04-0.01) |
| World Bank Low Income | Female | 0.09 (0.16-0.03) | 0.03 (0.06-0.01) |
| World Bank Lower Middle Income | Both sexes | 0.07 (0.12-0.02) | 0.02 (0.04-0.01) |
| World Bank Lower Middle Income | Male | 0.07 (0.11-0.02) | 0.05 (0.08-0.02) |
| World Bank Lower Middle Income | Female | 0.07 (0.13-0.02) | 0.05 (0.08-0.02) |
| World Bank Upper Middle Income | Both sexes | 0.12 (0.20-0.03) | 0.05 (0.07-0.03) |
| World Bank Upper Middle Income | Male | 0.11 (0.19-0.03) | 0.05 (0.08-0.02) |
| World Bank Upper Middle Income | Female | 0.13 (0.23-0.04) | 0.05 (0.08-0.02) |

ALL, Acute Lymphoblastic Leukemia; DALYs, disability-adjusted life-years; UI, uncertainty interval

projected to decline across all income subgroups within LMICs. The most substantial reductions are expected in low-income countries (LICs) (Fig 2).

## Discussion

Despite global medical advances, acute lymphoblastic leukemia (ALL) remains a significant health challenge for children and adolescents, particularly in low- and middle-income countries (LMICs) [9]. This study estimated the incidence, mortality, and disability-adjusted life years (DALYs) of ALL in LMICs across World Bank income groups, examined temporal trends from 1990 to 2023, and projected these trends to 2050.

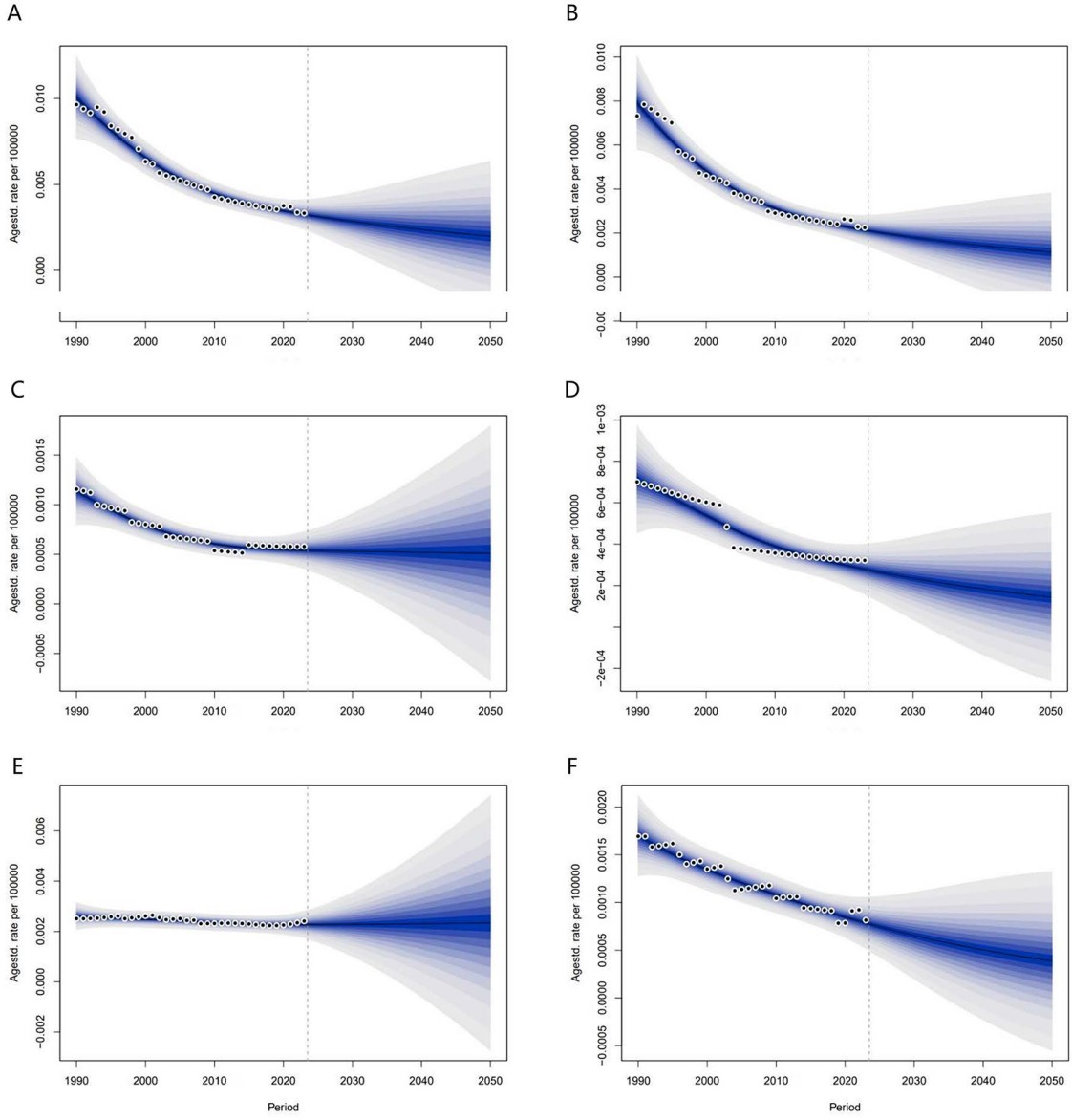

**Fig 2. Predictions for ASIR and ASMR of ALL in LMICs from 2023 to 2050.** LICs **(A and B)**; LMICs **(C and D)**; UMICs **(E and F)**.

Our study revealed a clear socioeconomic gradient in ALL burden. In 2023, age-standardized incidence, mortality, and DALY rates all increased progressively from low-income to upper-middle-income countries. This pattern likely reflects a combination of genuine risk differences and diagnostic capacity [10]. UMICs, as traditional high-incidence regions, may have higher genetic susceptibility and greater exposure to urbanization-related chemicals [11]. At the same time, they possess more robust cancer registration and diagnostic infrastructure, leading to more complete case detection. In contrast, LICs face severe resource constraints, low healthcare access, and likely underdiagnosis, meaning the true burden may be underestimated [12]. Males consistently bore a higher burden than females across all income groups. Age-specific patterns also differed by income level: in UMICs, ASIR peaked in infancy (<1 year) for both sexes, whereas in LICs and LMICs, the highest rates concentrated in the 2–4 years age group. This age shift may be explained by the delayed

infection hypothesis (Greaves' hypothesis): reduced early-life microbial exposure in UMICs delays ALL onset to infancy, whereas early infections in LICs/LMICs trigger earlier transformation at 2–4 years [13]. Diagnostic capacity also contributes—UMICs more readily detect aggressive infant ALL, while diagnostic delays in LICs/LMICs shift the apparent peak to older ages. Additionally, the highest incidence in children under 5 years reflects the global biological peak of ALL in the first five years of life, driven by immune development and hematopoietic proliferation windows [14]. Notably, although UMICs had the highest incidence rates, their mortality rates were relatively lower, suggesting better treatment access and survival outcomes. By contrast, LICs demonstrated a high mortality-to-incidence ratio, indicating that a large proportion of diagnosed cases result in death [15].

From 1990 to 2023, age-standardized incidence, mortality, and DALY rates declined across all LMIC subgroups, with the magnitude of decline increasing by national income level [16]. This pattern reflects socioeconomic progress, improved understanding of ALL pathogenesis, better identification of risk factors, advances in pediatric treatment protocols, and increased availability of diagnostic and treatment services [17]. The relatively modest declines in LICs likely reflect persistent resource constraints, uneven distribution of medical services, and limited application of advanced technologies [18,19]. The disease burden among infants (<1 year) remained relatively stable compared with other age groups, highlighting the persistent vulnerability of this population. Early diagnosis is crucial for improving survival [20], but translating diagnostic advances into population health benefits depends fundamentally on economic resources [21]. In UMICs, higher health investment facilitates integration of new biomarkers into screening programs, potentially accelerating future burden reductions [22,23]. In LICs, lack of laboratory diagnostic resources and risk stratification equipment hinders accurate diagnosis, even when effective tools exist, creating a "diagnostic gap" that exacerbates health inequalities [24]. We project that ASIR and ASMR for childhood and adolescent ALL will continue to decline across all LMIC subgroups from 2023 to 2050, with the largest reductions expected in LICs. However, these projections assume continuity of historical trends and may not capture sudden changes in healthcare policy or treatment breakthroughs [25].

This study has several limitations. First, although GBD 2023 provides standardized estimates, the quality of underlying data—such as cancer registration and cause-of-death systems—varies widely across countries. In regions with weaker health information systems, underreporting and misclassification may lead to underestimation of the true ALL burden. Second, our risk factor analysis is restricted to those included in the GBD framework. Other important etiologies, such as genetic susceptibility, viral infections, and other environmental exposures (e.g., pesticides, ionizing radiation), were not incorporated [26]. Third, the attribution of ALL burden to "occupational" benzene and formaldehyde exposure in children and adolescents warrants careful interpretation, as these estimates likely serve as proxies for child labor in informal sectors or para-occupational exposure from caregivers. This introduces potential exposure misclassification [27]. Finally, the BAPC models used for projections assume continuity of historical trends and do not account for potential major disruptions, such as treatment breakthroughs or sudden policy changes [28].

## Conclusions

Between 1990 and 2023, the burden of childhood and adolescent ALL in LMICs declined, but with a clear socioeconomic gradient. Improvements were smallest in low-income countries, where age-standardized DALY rates decreased only modestly or stagnated, highlighting persistent disparities in healthcare resources and access [29]. Males consistently bore a higher burden than females across all income groups. Notably, the gender gap narrowed in the poorest settings, likely due to universally poor survival rather than true equity in disease occurrence or treatment access.

Occupational benzene and formaldehyde exposure contributed negligibly to the total pediatric ALL burden (PAF < 1%), indicating that direct occupational exposure is rare in children. This finding does not negate the carcinogenicity of these chemicals but underscores that, from a population health perspective, they are not primary drivers of childhood ALL. Therefore, public health efforts should prioritize improving healthcare access, early diagnosis, and treatment outcomes rather than overemphasizing these specific environmental controls [29].

In summary, reducing the ALL burden in LMICs requires strengthening basic diagnostic and treatment capacity—particularly in low-income countries—rather than focusing on occupational exposures that account for only a minimal fraction of childhood disease. Closing the global survival gap will depend on sustained investment in accessible, high-quality care for all children, regardless of where they live [30].

## Supporting information

**S1 Table. Incidence, deaths, and DALYs number of acute lymphoblastic leukemia in 2023.**
(DOCX)

**S2 Table. Age-standardized incidence, mortality and DALYs rate of acute lymphoblastic leukemia in 2023.**
(DOCX)

**S3 Table. Average annual percent change of age-standardized incidence, mortality and DALYs rate from 1990 to 2023.**
(DOCX)

**S4 Table. Age-standardized DALYs rates for acute lymphoblastic leukemia attributable to risk factors in 2023.**
(DOCX)

**S5 Table. Age-standardized DALYs rate of average annual percent change attributable to risk factors for acute lymphoblastic leukemia from 1990 to 2023.**
(DOCX)

## Acknowledgments

We are grateful to the 2023 Global Burden of Disease Study collaborators for providing the data used in this study.

## Author contributions

**Data curation:** Peng Liu.

**Writing – original draft:** Peng Liu, ZiXin Xu.

**Writing – review & editing:** Wenfu Song, Jianxiong Yang, Jianbao Li.

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
