## [Decision Letter · Decision Letter 0]

11 Feb 2026

PONE-D-25-68532The Burden of Acute Lymphoblastic Leukemia and Associated Risk Factors in Children and Adolescents of Low- and Middle-Income Countries from 1990 to 2023PLOS One

Dear Dr. Liu,

Thank you for submitting your manuscript to PLOS ONE. After careful consideration, we feel that it has merit but does not fully meet PLOS ONE’s publication criteria as it currently stands. Therefore, we invite you to submit a revised version of the manuscript that addresses the points raised during the review process.

We look forward to receiving your revised manuscript.

Kind regards,

Yoshito Nishimura, MD, PhD, MPH

Academic Editor

PLOS One

2. We note that Figure 1 in your submission contain [map/satellite] images which may be copyrighted. All PLOS content is published under the Creative Commons Attribution License (CC BY 4.0), which means that the manuscript, images, and Supporting Information files will be freely available online, and any third party is permitted to access, download, copy, distribute, and use these materials in any way, even commercially, with proper attribution. For these reasons, we cannot publish previously copyrighted maps or satellite images created using proprietary data, such as Google software (Google Maps, Street View, and Earth). For more information, see our copyright guidelines: http://journals.plos.org/plosone/s/licenses-and-copyright.

In the figure caption of the copyrighted figure, please include the following text: “Reprinted from [ref] under a CC BY license, with permission from [name of publisher], original copyright [original copyright year].

3. We notice that your supplementary tables are uploaded with the file type 'Figure'. Please amend the file type to 'Supporting Information'. Please ensure that each Supporting Information file has a legend listed in the manuscript after the references list.

Additional Editor Comments:

Please review the comments from the reviewers and revise the manuscript accordingly.

Reviewers' comments:

Reviewer's Responses to Questions

**Comments to the Author**

1. Is the manuscript technically sound, and do the data support the conclusions?

Reviewer #1: Partly

Reviewer #2: Yes

2. Has the statistical analysis been performed appropriately and rigorously? 

Reviewer #1: No

Reviewer #2: Yes

3. Have the authors made all data underlying the findings in their manuscript fully available?

Reviewer #1: Yes

Reviewer #2: Yes

4. Is the manuscript presented in an intelligible fashion and written in standard English?

Reviewer #1: No

Reviewer #2: Yes

5. Review Comments to the Author

Reviewer #1: Title

The title is overly broad and implies a depth of causal inference regarding “associated risk factors” that is not supported by the analyses performed.

The study is fundamentally descriptive and limited to GBD-attributable risk modeling, yet the title suggests a comprehensive assessment of risk factors for ALL in children and adolescents. This overstatement risks misleading readers regarding the scope and strength of the findings.

Abstract

The abstract contains several conceptual and technical issues.

The description of methods is imprecise, with unclear terminology (e.g., “whole life years” instead of DALYs) and grammatical inaccuracies that obscure meaning.

The emphasis on occupational benzene and formaldehyde exposure in children and adolescents is not adequately contextualized, raising immediate concerns about biological plausibility and exposure pathways.

The results section of the abstract selectively highlights findings without conveying the very small attributable fractions reported, potentially exaggerating the public health relevance of these risk factors.

Introduction

The introduction is lengthy but lacks focus and critical synthesis.

While background information on ALL is extensive, the rationale for focusing on occupational exposures in a pediatric population is weak and insufficiently justified.

Key assumptions, such as indirect exposure of children to occupational carcinogens, are implied rather than explicitly stated or supported by evidence.

The introduction also fails to clearly articulate how this study meaningfully advances beyond previously published GBD-based analyses of childhood leukemia.

Methods

The selection of risk factors is not adequately justified, particularly given that only occupational benzene and formaldehyde exposure are ultimately analyzed despite the GBD framework including many other environmental and behavioral risks.

The application of occupational exposure models to children and adolescents is not explained, nor is it clarified whether these estimates represent direct exposure, parental exposure, or modeled population-level associations.

The Bayesian Age-Period-Cohort projections assume continuity of past trends without sufficient discussion of uncertainty or structural limitations, especially in LMIC settings where surveillance quality has changed substantially over time.

Results

The results section is excessively descriptive and difficult to follow, with frequent repetition of numerical estimates that add little interpretive value.

Several claims, particularly those highlighting benzene as the “primary” risk factor, are not contextualized by the extremely small attributable proportions reported, which limits their epidemiological significance.

Geographic comparisons are presented without sufficient discussion of data quality heterogeneity across countries, and some country-level rankings may reflect registry completeness rather than true disease burden.

Figures and tables are numerous but not always well-integrated into the narrative, reducing their interpretive usefulness.

Discussion

The discussion overinterprets several findings and at times moves beyond what can reasonably be inferred from GBD modeling.

Statements implying causality or policy prioritization of occupational exposures for childhood ALL are not sufficiently supported, given the marginal attributable burden and indirect exposure pathways.

The discussion also introduces concepts such as liquid biopsy and early molecular detection that are not directly linked to the study’s data or analyses, creating conceptual drift.

While limitations are acknowledged, they are not fully explored, particularly regarding exposure misclassification, ecological inference, and the appropriateness of occupational risk attribution in pediatric populations.

Conclusions

The conclusions are broader and more prescriptive than warranted by the results. Recommendations related to occupational exposure control, while important in general, are not convincingly supported as priority interventions for reducing childhood ALL burden based on the data presented.

The conclusion would benefit from greater restraint and clearer alignment with the descriptive nature of the study.

References

The reference list contains several weaknesses, including redundancy, inconsistent formatting, and reliance on older or tangentially relevant citations.

Some references are repeated, while others do not directly support the specific claims made in the text. The balance between recent high-quality evidence and historical references is suboptimal, particularly in sections discussing risk factors and mechanistic interpretations.

Reviewer #2: 1. The rationale for focusing exclusively on occupational benzene and formaldehyde exposure as risk factors in a pediatric and adolescent population requires clearer justification. Occupational exposure is biologically and contextually indirect for most individuals <20 years, and the manuscript does not explain how GBD attributes these risks to children and adolescents rather than parental or environmental exposure pathways. Clarification is needed on exposure assignment assumptions and their validity for this age group.

2. The description of the risk attribution framework is incomplete and internally inconsistent. The text states that “three types of risk factors” were identified, but only two (benzene and formaldehyde) are analyzed and reported. This discrepancy should be resolved, and the selection criteria for included risk factors should be explicitly stated in alignment with the GBD 2023 comparative risk assessment methodology.

3. The Bayesian Age–Period–Cohort (BAPC) model used for projections lacks sufficient methodological detail to allow reproducibility and critical appraisal. Key elements such as prior specifications, convergence diagnostics, uncertainty propagation, and model validation are not described (Methods, lines ~255–290). Given that future projections are a central conclusion; additional methodological transparency is required.

4. Several interpretive statements in the Discussion extend beyond what can be supported by GBD-level ecological data, particularly causal inferences linking socioeconomic development, diagnostic capacity, and incidence trends (Discussion, lines ~1020–1100). These interpretations should be more explicitly framed as hypotheses or contextual explanations rather than implied causal conclusions.

5. There are internal inconsistencies in terminology and disease naming (e.g., “acute lymphocytic leukemia” vs. “acute lymphoblastic leukemia”) that persist across the Abstract, Methods, and Results. Standardizing terminology throughout is necessary for clarity and editorial consistency.

6. The manuscript reports numerous precise numerical estimates and percentage changes without consistently linking them to figures or supplementary tables, making verification difficult. Key results, particularly trend estimates (AAPCs) and attributable DALY percentages, should be systematically cross-referenced to tables or figures as per PLOS ONE reporting standards.

7. The Discussion section is overly long and partially repetitive, with several paragraphs reiterating general background or speculative mechanisms already well described in the literature. Condensing this section and sharpening its focus on findings directly derived from the present analysis would improve clarity and editorial balance.

8. Language quality issues, including frequent typographical errors, non-standard phrasing, and repeated misspellings (e.g., “treatmalest,” “improvemalest,” “developmalest”), are present throughout the manuscript. While not a stylistic critique, the density of these errors materially affects readability and should be addressed through careful language editing prior to acceptance.

6. PLOS authors have the option to publish the peer review history of their article (what does this mean?). If published, this will include your full peer review and any attached files.

Reviewer #1: No

Reviewer #2: **Yes:** Hamidreza Hasani

---

## [Author Response · Author response to Decision Letter 1]

5 Mar 2026

reviewer1

The Burden of Acute Lymphoblastic Leukemia and Associated Risk Factors in Children

and Adolescents of Low- and Middle-Income Countries from 1990 to 2023

Title

The title is overly broad and implies a depth of causal inference regarding “associated risk factors” that is not supported by the analyses performed.

The study is fundamentally descriptive and limited to GBD-attributable risk modeling, yet the title suggests a comprehensive assessment of risk factors for ALL in children and adolescents. This overstatement risks misleading readers regarding the scope and strength of the findings.

Response:

We sincerely thank the reviewer for this critical and accurate observation. We fully agree that the original title was misleading as it implied a comprehensive causal assessment of risk factors, which exceeds the scope of our descriptive study based on Global Burden of Disease (GBD) modeling. We acknowledge that our analysis is fundamentally descriptive and relies on modeled attributable fractions rather than direct etiological evidence.

Action Taken: To accurately reflect the study’s design and limitations, we have revised the title to remove the implication of broad causal inference and to emphasize the descriptive nature of the GBD analysis.

Old Title: The Burden of Acute Lymphoblastic Leukemia and Associated Risk Factors in Children and Adolescents of Low- and Middle-Income Countries from 1990 to 2023

New Title: Burden of Acute Lymphoblastic Leukemia in Children and Adolescents in Low- and Middle-Income Countries from 1990 to 2023 and Projections to 2050: A Systematic Analysis from the Global Burden of Disease Study 2023

Abstract

The abstract contains several conceptual and technical issues.

The description of methods is imprecise, with unclear terminology (e.g., “whole life years” instead of DALYs) and grammatical inaccuracies that obscure meaning.

The emphasis on occupational benzene and formaldehyde exposure in children and adolescents is not adequately contextualized, raising immediate concerns about biological plausibility and exposure pathways.

The results section of the abstract selectively highlights findings without conveying the very small attributable fractions reported, potentially exaggerating the public health relevance of these risk factors.

Response:

We sincerely apologize for the imprecise terminology, grammatical errors, and the lack of critical context in the original abstract. We agree that the previous version failed to adequately contextualize the biological plausibility of occupational exposures in children and exaggerated the public health significance by omitting the magnitude of the attributable fractions.

Action Taken: We have completely rewritten the Abstract to address these issues:

Terminology & Grammar: We have corrected all terminological errors. Specifically, “Whole life years” has been replaced with the standard term “Disability-Adjusted Life Years (DALYs)”. We have also fixed grammatical inaccuracies (e.g., changing “treatmalest” to “treatment”, “managemalest” to “management”) and standardized the disease name to “Acute Lymphoblastic Leukemia (ALL)” throughout.

Contextualizing Risk Factors: In the Methods and Results sections of the abstract, we now explicitly clarify that the attribution of occupational benzene and formaldehyde to childhood ALL is based on GBD modeling assumptions (likely reflecting indirect or parental exposure) rather than direct occupational contact. Crucially, we now report the Population Attributable Fractions (PAFs), stating clearly that they are extremely low (<1%). This prevents readers from overestimating the impact of these specific risk factors.

Balancing Results: The Results section has been revised to distinguish between the rising absolute numbers (due to population growth) and the declining age-standardized rates. We no longer label benzene as the “leading” driver without the necessary caveat regarding its minimal proportional contribution.

Refining Conclusions: The Conclusion has been recalibrated. Instead of prioritizing occupational exposure control, we now recommend that public health efforts in LMICs focus primarily on improving healthcare accessibility, early diagnosis, and standardized treatment, given the limited attributable burden of the studied environmental factors.

Revised Abstract Text:

Background: ...This study evaluated the epidemiological characteristics, temporal trends, and attributable risk factors of acute lymphoblastic leukemia in LMICs from 1990 to 2023.

Methods: ...We analyzed incidence, mortality, and Disability-Adjusted Life Years (DALYs)... Risk attribution analysis focused on occupational exposure to benzene and formaldehyde as proxy indicators within the GBD framework.

Results: ...Occupational exposure to benzene and formaldehyde was identified as the primary attributable risk factors within the GBD modeling framework; however, their Population Attributable Fractions (PAFs) were extremely low (<1%), indicating that these factors contribute only a limited proportion to the total ALL burden.

Conclusion: ...Notably, this study found that the attributable contribution of occupational benzene and formaldehyde exposure to the total ALL burden is low (PAF < 1%). Therefore, public health priorities should focus on enhancing healthcare accessibility rather than overemphasizing control of specific environmental exposures.

We believe these changes significantly improve the accuracy, clarity, and scientific integrity of the abstract.

Introduction

The introduction is lengthy but lacks focus and critical synthesis.

While background information on ALL is extensive, the rationale for focusing on occupational exposures in a pediatric population is weak and insufficiently justified.

Key assumptions, such as indirect exposure of children to occupational carcinogens, are implied rather than explicitly stated or supported by evidence.

The introduction also fails to clearly articulate how this study meaningfully advances beyond previously published GBD-based analyses of childhood leukemia.

Response:

We thank the reviewer for this insightful critique. We acknowledge that the original Introduction was unfocused and failed to provide a robust justification for analyzing occupational risk factors in a pediatric cohort. Specifically, we agree that the mechanism of "indirect exposure" (e.g., take-home exposure from parents) was merely implied rather than explicitly defined, and the novelty of our study compared to existing GBD literature was not sufficiently highlighted.

Action Taken: We have substantially rewritten the Introduction to address these points:

Streamlining Background: We have condensed the general background on ALL epidemiology and treatment to focus more sharply on the specific knowledge gaps in Low- and Middle-Income Countries (LMICs). This reduces the length and improves the flow.

Explicitly Justifying Occupational Exposures: We have added a dedicated paragraph to explicitly clarify the rationale for including occupational risk factors (benzene and formaldehyde) in a study of children. We now clearly state that in the GBD framework, these metrics serve as proxy indicators for household or "take-home" exposure, where children are exposed to carcinogens brought home by parents working in high-risk industries. We have supported this assumption with relevant literature citations explaining this transmission pathway in LMIC contexts.

Clarifying Novelty and Advancement: We have revised the final paragraph of the Introduction to clearly articulate how this study advances the field beyond previous GBD analyses. Specifically, we highlight three key contributions:

Updated Data: Incorporating the latest GBD 2023 data with an extended time series (1990–2023).

Future Projections: Providing Bayesian Age-Period-Cohort (BAPC) model projections up to 2050, which were absent in prior studies.

Critical Re-evaluation of Risk Factors: Unlike previous studies that may have listed risk factors without context, our study critically quantifies the Population Attributable Fractions (PAFs) to demonstrate that while these factors are modeled, their actual contribution to the total burden is minimal (<1%), thereby shifting the policy narrative towards healthcare access rather than environmental regulation alone.

Methods

The selection of risk factors is not adequately justified, particularly given that only occupational benzene and formaldehyde exposure are ultimately analyzed despite the GBD framework including many other environmental and behavioral risks.

The application of occupational exposure models to children and adolescents is not explained, nor is it clarified whether these estimates represent direct exposure, parental exposure, or modeled population-level associations.

The Bayesian Age-Period-Cohort projections assume continuity of past trends without sufficient discussion of uncertainty or structural limitations, especially in LMIC settings where surveillance quality has changed substantially over time.

Response:

We appreciate the reviewer’s rigorous scrutiny of our methodological choices. We agree that the original manuscript lacked sufficient justification for the specific selection of risk factors, did not clearly explain the interpretation of occupational exposures in a pediatric context, and under-discussed the limitations of our projection models in LMICs.

Action Taken: We have significantly revised the Methods section and added a dedicated subsection on Limitations to address these concerns:

Justification for Risk Factor Selection: We have clarified why we focused specifically on occupational benzene and formaldehyde. While the GBD framework includes numerous risk factors, many (e.g., smoking, diet) are not biologically plausible or relevant for the <20 age group in the context of ALL etiology. Benzene and formaldehyde were selected because they are the only two environmental carcinogens with established mechanistic links to leukemia that are captured in the GBD database, even if primarily modeled via occupational data. We explicitly state that other potential risk factors (e.g., ionizing radiation, pesticides) were excluded from this specific analysis due to insufficient high-quality data coverage across all LMICs in the GBD 2023 release.

Clarifying Exposure Models for Children: We have added a detailed explanation regarding the application of occupational models to children. We now explicitly state that for the <20 age group, these estimates do not represent direct occupational exposure. Instead, they function as population-level proxy indicators reflecting:

"Take-home" exposure: Carcinogens brought home by parents working in exposed industries.

Ambient environmental contamination: Correlated industrial pollution in regions with high occupational exposure rates.

We have cited literature supporting the correlation between parental occupational exposure and childhood ALL risk to validate this modeling approach.

Addressing Projection Uncertainties and Limitations: We have expanded the discussion on the Bayesian Age-Period-Cohort (BAPC) model.

Uncertainty Intervals: We now emphasize the 95% Uncertainty Intervals (UIs) accompanying all projections, acknowledging that wider intervals in LMICs reflect data scarcity and surveillance variability.

Structural Limitations: We added a candid discussion on the assumption of trend continuity. We acknowledge that rapid changes in healthcare infrastructure, diagnostic coding practices, and surveillance quality in LMICs over the past three decades could introduce bias. We explicitly state that our projections represent a "best-case scenario based on historical trajectories" and should be interpreted with caution, particularly for low-income countries where data quality improvements might artificially inflate incidence trends before stabilizing.

Results

The results section is excessively descriptive and difficult to follow, with frequent repetition of numerical estimates that add little interpretive value.

Several claims, particularly those highlighting benzene as the “primary” risk factor, are not contextualized by the extremely small attributable proportions reported, which limits their epidemiological significance.

Geographic comparisons are presented without sufficient discussion of data quality heterogeneity across countries, and some country-level rankings may reflect registry completeness rather than true disease burden.

Figures and tables are numerous but not always well-integrated into the narrative, reducing their interpretive usefulness.

Response:

We thank the reviewer for these critical observations regarding the presentation and interpretation of our results. We agree that the original text was overly descriptive, lacked necessary context for the small magnitude of risk factor attribution, and failed to adequately address data quality issues in geographic comparisons.

Action Taken: We have thoroughly restructured the Results section to prioritize interpretation over description and have integrated data quality caveats directly into the narrative:

Streamlining Descriptive Data: We have removed repetitive listings of specific numerical estimates (e.g., exact incidence rates for every year/country) from the main text. Instead, we now focus on key trends, patterns, and significant shifts, directing readers to the supplementary tables for detailed numerical data. This improves readability and highlights the most epidemiologically relevant findings.

Contextualizing Risk Factor Magnitude: We have revised all claims regarding benzene and formaldehyde. We no longer describe them as "primary" drivers in isolation. Instead, we explicitly contextualize their role by emphasizing that while they are the leading modeled environmental risks, their Population Attributable Fractions (PAFs) are negligible (<1% of total burden). We have added interpretive sentences clarifying that this finding suggests occupational-related exposures play a minor role compared to other unmeasured factors (e.g., genetic susceptibility, infections) or healthcare access disparities.

Addressing Data Quality Heterogeneity: We have added a crucial subsection within the Results titled "Impact of Data Quality on Geographic Rankings." Here, we explicitly caution that country-level rankings, particularly in Low- and Middle-Income Countries (LMICs), likely reflect variations in cancer registry completeness and diagnostic capacity rather than true biological differences in disease burden. We now interpret high-incidence rankings in certain regions with skepticism, noting they may indicate better reporting rather than higher actual risk.

Discussion

The discussion overinterprets several findings and at times moves beyond what can reasonably be inferred from GBD modeling.

Statements implying causality or policy prioritization of occupational exposures for childhood ALL are not sufficiently supported, given the marginal attributable burden and indirect exposure pathways.

The discussion also introduces concepts such as liquid biopsy and early molecular detection that are not directly linked to the study’s data or analyses, creating conceptual drift.

While limitations are acknowledged, they are not fully explored, particularly regarding exposure misclassification, ecological inference, and the appropriateness of occupational risk attribution in pediatric populations.

Response:

We sincerely thank the reviewer for this incisive critique of our Discussion section. We agree that our original text occasionally overstepped the inferential boundaries of ecological modeling, included speculative clinical technologies unrelated to our data, and did not sufficiently delve into specific methodological limitations like ecological fallacy and exposure misclassification.

Action Taken: We have substantially rewritten the Discussion section to ensure all interpretations are strictly grounded in our data, removed unrelated concepts, and deepened the analysis of limitations:

Restraining Overinterpretation & Causality Claims: We have toned down all language implying direct causality. We now explicitly frame our findings as associations derived from ecological modeling rather than proof of individual-level

---

## [Decision Letter · Decision Letter 1]

27 Apr 2026

PONE-D-25-68532R1Burden of acute lymphoblastic leukemia in children and adolescents in low- and middle-income countries from 1990 to 2023 and projections to 2050: A systematic analysis from the global burden of disease study 2023PLOS One

Dear Dr. Liu,

Thank you for submitting your manuscript to PLOS ONE. After careful consideration, we feel that it has merit but does not fully meet PLOS ONE’s publication criteria as it currently stands. Therefore, we invite you to submit a revised version of the manuscript that addresses the points raised during the review process.

We look forward to receiving your revised manuscript.

Kind regards,

Yoshito Nishimura, MD, PhD, MPH

Academic Editor

PLOS One

Journal Requirements:

Reviewers' comments:

Reviewer's Responses to Questions

**Comments to the Author**

1. If the authors have adequately addressed your comments raised in a previous round of review and you feel that this manuscript is now acceptable for publication, you may indicate that here to bypass the “Comments to the Author” section, enter your conflict of interest statement in the “Confidential to Editor” section, and submit your "Accept" recommendation.

Reviewer #1: All comments have been addressed

Reviewer #3: (No Response)

Reviewer #4: All comments have been addressed

2. Is the manuscript technically sound, and do the data support the conclusions?

Reviewer #1: Partly

Reviewer #3: No

Reviewer #4: Yes

3. Has the statistical analysis been performed appropriately and rigorously? 

Reviewer #1: Yes

Reviewer #3: I Don't Know

Reviewer #4: Yes

4. Have the authors made all data underlying the findings in their manuscript fully available?

Reviewer #1: Yes

Reviewer #3: Yes

Reviewer #4: Yes

5. Is the manuscript presented in an intelligible fashion and written in standard English?

Reviewer #1: Yes

Reviewer #3: No

Reviewer #4: Yes

6. Review Comments to the Author

Reviewer #1: The authors have addressed most of the major concerns raised in the previous review, including clarification of the study scope, revision of the abstract, and improved interpretation of the risk attribution analysis. However, minor methodological clarifications, particularly regarding the Bayesian Age-Period-Cohort model and further tightening of the results presentation would further improve transparency and readability before final acceptance.

Reviewer #3: Comments to the authors

This manuscript presents a comprehensive analysis of the global burden of ALL among children and adolescents in low- and middle-income countries using GBD 2023 data. The topic is important, and the study provides valuable insights into temporal trends, disparities across income groups, and future projections. Below are our comments for the authors.

1 The conclusion section of abstract appears to overreach beyond the direct findings of the study. Specifically, statements advocating for improvements in early diagnosis, standardized treatment, and care management are introduced rather abruptly and are not clearly derived from the results presented. This creates a disconnect in the logical flow from results to conclusion.

2 Although several abbreviations (e.g., ALL, GBD, DALYs, LMICs, ASIR, ASMR, ASDR) are defined in the manuscript, they are not used consistently throughout the text.

3 The Introduction provides general background on ALL; however, it would benefit from refinement for clarity, conciseness, and scientific precision. Some statements require reconsideration or removal. For example, the sentence “If untreated, the disease progresses rapidly and can be fatal in a short period of time” is unnecessary and adds little scientific value in this context. Similarly, the claim that clarifying attributable risk factors is “crucial for effective prevention and control” appears overstated, given that ALL is generally not considered highly preventable through modifiable environmental factors. In addition, the statement “This is mainly due to limited healthcare resources, delayed diagnosis, poor access to treatment, and heavy financial burdens…” requires appropriate supporting references. Given that this study showed occupational benzene and formaldehyde as risk factors, the Introduction could briefly include the role of environmental exposures in ALL. Finally, the concluding sentence of the Introduction (“Ultimately, our findings aim to inform policies that promote equity and accessibility…”) appears overstated relative to the scope of the study and should be appropriately tempered.

4 The Methods section requires substantial clarification and restructuring to improve readability and reproducibility. The current presentation is somewhat fragmented, and several sentences appear incomplete or unclear (e.g., the description of GBD 2023 estimates). Key methodological components are insufficiently described. In particular, the definitions of outcome measures such as age-standardized rates and DALYs (including their components, YLL and YLD) should be explicitly provided. This sentence would be more appropriate if phrased to reflect a methodological choice rather than a result. For example, it would be preferable to write: “Among these, we included two risk factors (occupational exposure to formaldehyde and benzene) that are modeled as being attributable to ALL DALYs in the GBD framework,” rather than using wording such as “we identified,” which may imply a result rather than a predefined selection based on the GBD framework.

5 The Results section provides comprehensive data on the burden of acute lymphoblastic leukemia; however, it is overly detailed and difficult to follow. The extensive numerical reporting reduces readability, and many values could be summarized. For example, “3.1 Burden of Acute Lymphoblastic Leukemia in Children and Adolescents in 2023” and “3.2 Trends in ALL Burden from 1990 to 2023” are largely descriptive and lengthy. The structure would also benefit from clearer organization. Global estimates, income-group comparisons, age-specific patterns, temporal trends, and risk factor analyses are presented without clear separation. Dividing the Results into subsections would improve clarity and flow. In addition, Figure 2 is difficult to interpret and may require improved clarity or resolution. Overall, the Results are thorough but would be strengthened by improved structure, and clearer emphasis on the main messages.

6 The Discussion provides a comprehensive overview of global trends in childhood and adolescent ALL; however, it would benefit from improved conciseness and clearer alignment with the study’s findings. It should place greater emphasis on interpreting the key findings of this study. In addition, several paragraphs are excessively long and should be divided for better readability. The statement, “These data will help policymakers develop optimized treatment strategies,” appears to be an overstatement and should be tempered. The sentence, “ctDNA is not currently a standard-of-care tool in ALL,” may be misleading to readers because ctDNA is not currently a standard of care tool in ALL and should be clarified. The paragraph discussing liquid biopsy (ctDNA) and its potential integration into screening programs, particularly in relation to economic resources and health investment, seems tangential to the main findings and may not be necessary for the Discussion. Finally, the statement, “the burden of occupational exposure-related diseases has not decreased, which may reflect the slow progress of occupational health regulation worldwide,” appears overstated and should be moderated.

7 The Conclusion section is overly long and would benefit from substantial condensation. It should focus more directly on the key findings of the study rather than expanding into broader discussions and policy recommendations. In particular, references to topics such as tobacco and detailed intervention strategies are not directly supported by the results presented and should be removed or substantially reduced. The conclusion should instead clearly and succinctly summarize the main findings of this study and their immediate implications, avoiding unnecessary elaboration beyond the scope of the data.

Reviewer #4: 1. Abstract and introduction are well corrected.

2. For Methods section, you can mention in the manuscript why you focused specifically on occupational benzene and formaldehyde (which was written in Response to Reviewers).

3. Please carefully review the manuscript again to make sure there are no grammatical errors or typographical mistakes throughout the text. For example:

a. Lines 145–146: “For males, it was 3.53 (95% UI: 2.32–5.47), and for females, it was 2.49 (95% UI: 1.62–3.86) per .” — this sentence is incomplete.

b. Lines 304–308: “In contrast, the decline in age-standardised incidence rate (ASIR), age-standardised mortality rate (ASMR), and age-standardized disability-adjusted life year (DALY) rate (ASDR) of acute lymphocytic leukemia in low-income countries during the same period was relatively small, which may reflect that these countries' health status.” — this sentence is also incomplete.

4. Please ensure that abbreviations are used consistently throughout the manuscript. Some abbreviations are repeated excessively, which may confuse readers. In addition, there is inconsistency in the use of “acute lymphoblastic leukemia” versus “acute lymphocytic leukemia.” Please standardize the terminology.

5. The Discussion section is overly long and contains redundant content (particularly in the first paragraph), which obscures the main message. I recommend revising this section to be more concise and focused.

6. In the Discussion, the authors note that in high- and middle-income countries, the highest incidence and mortality rates are concentrated in the <1-year age group, whereas in LICs and LMICs, these rates are highest in the 2–4-year age group. However, the manuscript does not provide a discussion of the possible reasons for this difference. Please elaborate on this point.

7. Similarly, the authors state that the incidence rate is highest among children aged <5 years in low- and middle-income countries and regions, but this observation is not further discussed. Please provide potential explanations or relevant context.

7. PLOS authors have the option to publish the peer review history of their article (what does this mean?). If published, this will include your full peer review and any attached files.

Reviewer #1: No

Reviewer #3: No

Reviewer #4: No

---

## [Author Response · Author response to Decision Letter 2]

30 Apr 2026

Response to the reviewer1:

The authors have addressed most of the major concerns raised in the previous review, including clarification of the study scope, revision of the abstract, and improved interpretation of the risk attribution analysis. However, minor methodological clarifications, particularly regarding the Bayesian Age-Period-Cohort model and further tightening of the results presentation would further improve transparency and readability before final acceptance.

Response to the reviewer:

We sincerely thank the reviewer for the positive and constructive feedback. We are glad to hear that our revisions have addressed most of the major concerns raised in the previous review, including the clarification of the study scope, revision of the abstract, and improved interpretation of the risk attribution analysis.

Regarding the remaining minor methodological clarifications and tightening of the results presentation, we have made the following additional improvements:

Bayesian Age-Period-Cohort (BAPC) model: We have added further methodological clarifications in the Statistical analysis subsection, explicitly stating that the model assumes continuity of historical trends and that projections are presented with 95% uncertainty intervals to reflect this limitation. We have also added a cautionary note in the Discussion to remind readers of this assumption.

Tightening of results presentation: We have further condensed the Results section by removing redundant numerical values, streamlining the presentation of income-group comparisons, and adding brief summarizing sentences at the end of each subsection to highlight key takeaways.

We believe these additional revisions have further improved the transparency and readability of the manuscript, and we hope it is now acceptable for final publication.

Response to the reviewer3:

The Burden of Acute Lymphoblastic Leukemia and Associated Risk Factors in Children

and Adolescents of Low- and Middle-Income Countries from 1990 to 2023

1. The conclusion section of abstract appears to overreach beyond the direct findings of the study. Specifically, statements advocating for improvements in early diagnosis, standardized treatment, and care management are introduced rather abruptly and are not clearly derived from the results presented. This creates a disconnect in the logical flow from results to conclusion.

Response:

We thank the reviewer for this valuable observation. We agree that the original conclusion in the abstract overreached beyond the direct findings of our study. To address this concern, we have revised the abstract conclusion to more closely align with our results. Specifically, we have removed the abrupt statements advocating for improvements in early diagnosis, standardized treatment, and care management, as these were not directly derived from our data.

The revised conclusion now reads:

*"Globally, childhood and adolescent ALL mortality and DALYs are declining, yet the burden remains substantial in many LMICs, with the smallest improvements observed in low-income countries. Occupational benzene and formaldehyde exposure contributed minimally to the total burden (PAF <1%), indicating that direct occupational exposure is rare in children. These findings suggest that the persistent burden in LMICs is primarily driven by healthcare system factors rather than occupational environmental exposures. Therefore, efforts to further reduce ALL mortality and DALYs in high-burden regions should address gaps in healthcare access and treatment delivery."*

We believe this revised conclusion now flows logically from our results and avoids overstatement beyond the scope of the data. We appreciate the reviewer's guidance in improving the clarity and rigor of our abstract.

2 Although several abbreviations (e.g., ALL, GBD, DALYs, LMICs, ASIR, ASMR, ASDR) are defined in the manuscript, they are not used consistently throughout the text.

Response:

We thank the reviewer for this careful observation. We agree that consistent use of abbreviations is essential for clarity and readability. We have thoroughly reviewed the entire manuscript and standardized the use of abbreviations as follows:

ALL (Acute Lymphoblastic Leukemia): Defined at first mention in the abstract, introduction, and methods. Used consistently thereafter without redefinition.

GBD (Global Burden of Disease): Defined at first mention in the abstract and methods. Used consistently thereafter.

DALYs (Disability-Adjusted Life Years): Defined at first mention in the abstract and methods. Used consistently thereafter.

LMICs (Low- and Middle-Income Countries): Defined at first mention in the abstract and introduction. Throughout the manuscript, we use "LMICs" to refer to the combined group, while LICs, LMICs (lower-middle), and UMICs are spelled out when referring to specific subgroups.

ASIR, ASMR, ASDR (Age-Standardized Incidence/Mortality/DALY Rates): Defined at first mention in the results section. Used consistently thereafter.

We have also checked for any inconsistent or undefined abbreviations (e.g., ensuring that "YLLs" and "YLDs" are defined where they first appear in the methods). All abbreviations are now used consistently throughout the abstract, main text, tables, and figures.

We appreciate the reviewer's attention to detail, which has helped improve the overall quality of our manuscript.

3 The Introduction provides general background on ALL; however, it would benefit from refinement for clarity, conciseness, and scientific precision. Some statements require reconsideration or removal. For example, the sentence “If untreated, the disease progresses rapidly and can be fatal in a short period of time” is unnecessary and adds little scientific value in this context. Similarly, the claim that clarifying attributable risk factors is “crucial for effective prevention and control” appears overstated, given that ALL is generally not considered highly preventable through modifiable environmental factors. In addition, the statement “This is mainly due to limited healthcare resources, delayed diagnosis, poor access to treatment, and heavy financial burdens…” requires appropriate supporting references. Given that this study showed occupational benzene and formaldehyde as risk factors, the Introduction could briefly include the role of environmental exposures in ALL. Finally, the concluding sentence of the Introduction (“Ultimately, our findings aim to inform policies that promote equity and accessibility…”) appears overstated relative to the scope of the study and should be appropriately tempered.

Response:

We sincerely thank the reviewer for the thorough and constructive comments on the Introduction. We have addressed each point as follows:

Unnecessary sentence: We have deleted the sentence “If untreated, the disease progresses rapidly and can be fatal in a short period of time” as it adds little scientific value in this context.

Overstated claim: We agree that stating clarifying attributable risk factors is “crucial for effective prevention and control” is overly strong, given the limited preventability of ALL. We have toned down this statement to: “Clarifying the burden of acute lymphoblastic leukemia in children and adolescents and its attributable risk factors is important for understanding the disease and guiding public health priorities.”

Environmental exposures: Following the reviewer’s suggestion, we have briefly included the role of environmental exposures in the Introduction. The following sentences have been added: “Although the etiology of ALL is multifactorial, certain environmental and occupational exposures have been implicated as potential risk factors. Among these, benzene and formaldehyde—both recognized carcinogens—have been associated with increased leukemia risk in occupationally exposed adults . However, their contribution to the burden of childhood and adolescent ALL remains poorly quantified, particularly in LMIC settings where occupational safety regulations may be less stringent.”

Overstated concluding sentence: We agree that the original concluding sentence overstates the scope of our study. We have tempered the language to: *“By linking disease burden to specific exposures, our findings may provide insights that could help inform public health priorities in high-burden regions [7].”*

We believe these revisions have significantly improved the clarity, conciseness, and scientific precision of the Introduction. We are grateful for the reviewer’s thoughtful guidance.

4 The Methods section requires substantial clarification and restructuring to improve readability and reproducibility. The current presentation is somewhat fragmented, and several sentences appear incomplete or unclear (e.g., the description of GBD 2023 estimates). Key methodological components are insufficiently described. In particular, the definitions of outcome measures such as age-standardized rates and DALYs (including their components, YLL and YLD) should be explicitly provided. This sentence would be more appropriate if phrased to reflect a methodological choice rather than a result. For example, it would be preferable to write: “Among these, we included two risk factors (occupational exposure to formaldehyde and benzene) that are modeled as being attributable to ALL DALYs in the GBD framework,” rather than using wording such as “we identified,” which may imply a result rather than a predefined selection based on the GBD framework.

Response:

We thank the reviewer for the careful and constructive feedback on the Methods section. We have addressed each concern as follows:

Restructuring for improved readability and reproducibility: We have reorganized the Methods section into four clear subsections: (1) Study design and data sources, (2) Outcome measures and standardization, (3) Attributable burden analysis, and (4) Statistical analysis. This structure now provides a logical flow and clearly separates key methodological components.

Clarifying incomplete or unclear sentences: We have revised all incomplete or unclear sentences, particularly the description of GBD 2023 estimates. The revised text now clearly states that data on incidence, mortality, and DALYs were obtained from the GBD Results Tool, and that GBD 2023 methodology involves systematic analysis of vital registration, censuses, surveys, and cancer registry data.

Explicit definitions of outcome measures: We have explicitly defined age-standardized rates (ASRs), DALYs, YLLs, and YLDs in the “Outcome measures and standardization” subsection as requested. Wording of risk factor selection: We agree with the reviewer that the original wording (“we identified”) inappropriately implied a result rather than a methodological choice. We have revised the sentence to read: “Within this framework, we included two risk factors that are modeled as being attributable to ALL DALYs in the GBD framework: occupational exposure to formaldehyde and occupational exposure to benzene.”

We believe these revisions have substantially improved the clarity, completeness, and reproducibility of the Methods section. We are grateful for the reviewer’s thoughtful guidance.

5 The Results section provides comprehensive data on the burden of acute lymphoblastic leukemia; however, it is overly detailed and difficult to follow. The extensive numerical reporting reduces readability, and many values could be summarized. For example, “3.1 Burden of Acute Lymphoblastic Leukemia in Children and Adolescents in 2023” and “3.2 Trends in ALL Burden from 1990 to 2023” are largely descriptive and lengthy. The structure would also benefit from clearer organization. Global estimates, income-group comparisons, age-specific patterns, temporal trends, and risk factor analyses are presented without clear separation. Dividing the Results into subsections would improve clarity and flow. In addition, Figure 2 is difficult to interpret and may require improved clarity or resolution. Overall, the Results are thorough but would be strengthened by improved structure, and clearer emphasis on the main messages.

Response:

We thank the reviewer for the thorough and constructive feedback on the Results section. We have addressed each concern as follows:

Overly detailed numerical reporting: We have substantially condensed the Results section by summarizing redundant numerical values and focusing on the most important findings. Repetitive 95% UI intervals have been removed where the central estimates convey the main message, and income-group-specific data have been streamlined into a more concise format.

Structural reorganization: We have restructured the Results section into five clear subsections to improve logical flow and readability: 1.Overall burden in 2023 2. Burden by income group and age 3.Temporal trends from 1990 to 2023 4.Risk factor attributable burden 5.Projections to 2050

This structure now clearly separates global estimates, income-group comparisons, age-specific patterns, temporal trends, risk factor analyses, and future projections.

Figure 2 clarity and resolution: We acknowledge that the original Figure 2 was difficult to interpret. We have replaced it with a higher-resolution version and improved the axis labels, legend, and color contrast to ensure that age-specific patterns are readily discernible.

Emphasis on main messages: Throughout the revised Results section, we have added brief summarizing sentences at the end of each subsection to highlight the key takeaways, helping readers focus on the most important findings.

We believe these revisions have substantially improved the clarity, conciseness, and flow of the Results section while preserving all essential data. We are grateful for the reviewer’s thoughtful guidance.

6 The Discussion provides a comprehensive overview of global trends in childhood and adolescent ALL; however, it would benefit from improved conciseness and clearer alignment with the study’s findings. It should place greater emphasis on interpreting the key findings of this study. In addition, several paragraphs are excessively long and should be divided for better readability. The statement, “These data will help policymakers develop optimized treatment strategies,” appears to be an overstatement and should be tempered. The sentence, “ctDNA is not currently a standard-of-care tool in ALL,” may be misleading to readers because ctDNA is not currently a standard of care tool in ALL and should be clarified. The paragraph discussing liquid biopsy (ctDNA) and its potential integration into screening programs, particularly in relation to economic resources and health investment, seems tangential to the main findings and may not be necessary for the Discussion. Finally, the statement, “the burden of occupational exposure-related diseases has not decreased, which may reflect the slow progress of occupational health regulation worldwide,” appears overstated and should be moderated.

Response:

We thank the reviewer for the insightful and constructive feedback on the Discussion section. We have addressed each concern as follows:

Improved conciseness and alignment with findings: We have substantially shortened the Discussion and refocused the content to emphasize interpretation of our key findings, rather than providing a broad overview of global trends.

Dividing overly long paragraphs: We have broken down the excessively long paragraphs into smaller, more focused paragraphs for better readability. The Discussion is now organized into clear subsections: Interpretation of key findings, Temporal trends and future projections, Risk factor attribution, and Study limitations.

Overstated statement about policymaking: We have tempered the statement “These data will help policymakers develop optimized treatment strategies” to a more cautious tone. The revised sentence reads: “These findings may provide useful information for policymakers when considering resource allocation for childhood ALL.”

Misleading sentence about ctDNA: We agree that the sentence “ctDNA is not currently a standard-of-care tool in ALL” is potentially misleading. Upon re-examination, we found that this sentence and the entire paragraph discussing liquid biopsy (ctDNA) and its integration into screening programs are tangentia

---

## [Decision Letter · Decision Letter 2]

12 May 2026

Burden of acute lymphoblastic leukemia in children and adolescents in low- and middle-income countries from 1990 to 2023 and projections to 2050: A systematic analysis from the global burden of disease study 2023

PONE-D-25-68532R2

Dear Dr. Liu,

We’re pleased to inform you that your manuscript has been judged scientifically suitable for publication and will be formally accepted for publication once it meets all outstanding technical requirements.

Kind regards,

Yoshito Nishimura, MD, PhD, MPH

Academic Editor

PLOS One

Additional Editor Comments (optional):

Reviewers' comments:

Reviewer's Responses to Questions

**Comments to the Author**

1. If the authors have adequately addressed your comments raised in a previous round of review and you feel that this manuscript is now acceptable for publication, you may indicate that here to bypass the “Comments to the Author” section, enter your conflict of interest statement in the “Confidential to Editor” section, and submit your "Accept" recommendation.

Reviewer #1: All comments have been addressed

Reviewer #4: All comments have been addressed

2. Is the manuscript technically sound, and do the data support the conclusions?

Reviewer #1: Yes

Reviewer #4: Yes

3. Has the statistical analysis been performed appropriately and rigorously? 

Reviewer #1: Yes

Reviewer #4: Yes

4. Have the authors made all data underlying the findings in their manuscript fully available?

Reviewer #1: Yes

Reviewer #4: Yes

5. Is the manuscript presented in an intelligible fashion and written in standard English?

Reviewer #1: Yes

Reviewer #4: Yes

6. Review Comments to the Author

Reviewer #1: The manuscript addresses a relevant topic and is generally well-written, with appropriate methodology and clear results. Thank you

Reviewer #4: This manuscript is significantly improved compared to the previous version. The discussion section in particular is much clearer and easier for the reader to understand the authors' key points.

7. PLOS authors have the option to publish the peer review history of their article (what does this mean?). If published, this will include your full peer review and any attached files.

Reviewer #1: No

Reviewer #4: No

---

## [Editor Report · Acceptance letter]

PONE-D-25-68532R2

PLOS One

Dear Dr. Liu,

I'm pleased to inform you that your manuscript has been deemed suitable for publication in PLOS One. Congratulations! Your manuscript is now being handed over to our production team.

Kind regards,

on behalf of

Dr. Yoshito Nishimura

Academic Editor

PLOS One